# Comparing Ant Assemblages and Functional Groups across Urban Habitats and Seasons in an East Asia Monsoon Climate Area

**DOI:** 10.3390/ani13010040

**Published:** 2022-12-22

**Authors:** Xin-Yu Luo, Chris Newman, Yi Luo, Zhao-Min Zhou

**Affiliations:** 1Key Laboratory of Southwest China Wildlife Resources Conservation (Ministry of Education), China West Normal University, Nanchong 637009, China; 2Wildlife Conservation Research Unit, Department of Biology, The Recanati-Kaplan Centre, University of Oxford, Oxford OX13 5QL, UK; 3Key Laboratory of Environmental Science and Biodiversity Conservation (Sichuan Province), China West Normal University, Nanchong 637009, China

**Keywords:** ant, assemblage composition, cropland, diversity, habitat restoration, landscape change, species richness, urban

## Abstract

**Simple Summary:**

Urban green spaces often provide refuges for native biodiversity. In light of human population growth and climate change, this research’s aim was to reveal how ant community structures differed among human-modified habitats. Ant diversity was sensitive to vegetation composition, habitat fragmentation, and seasonal change, while ant functional groups responded differently to these factors, which further enhanced their sensitivity. Therefore, future conservation and monitoring plans for ant diversity should take seasonality, habitat filters, and functional groups into greater consideration.

**Abstract:**

China’s East Asia monsoon zone is undergoing rapid land-use conversion and urbanization. Safeguarding remaining biodiversity requires reducing, mitigating, and/or eliminating the negative impacts of human-induced landscape modification. In this study, we sampled ground-dwelling ants at 40 plots over 12 continuous months in a suburban area in southwestern China to examine whether and how vegetation composition and habitat fragmentation affected species richness and assemblage composition for the general ant community and, specifically, for principal functional groups (including Opportunists and Generalized Myrmicinae). Warmer seasons were associated with a higher capture rate for all functional groups. Patterns of ant species richness among Opportunists were more sensitive to vegetation and fragmentation than for Generalized Myrmicinae, and these effects generally varied with season. Patterns of ant assemblage composition for Opportunists were exclusively sensitive to vegetation, whereas Generalized Myrmicinae were sensitive to both vegetation and fragmentation with variation among seasons. Overall, our findings highlight the important role of seasonality, vegetation composition, and habitat fragmentation in mediating the impacts of human-induced landscape modification on urbanized ant communities, which make an essential functional contribution to biodiversity in the East Asia monsoon zone.

## 1. Introduction

Urbanization is an increasingly important driver of global change [1,2,3,4] that is causing worldwide declines in the abundance, diversity, and health of species and ecosystems, which makes it one of the greatest contemporary threats to nature [5,6,7,8]. This is a particular issue across the East Asia monsoon zone (latitude 20–40° N, longitude 100–145° E). Here, some 3.4 billion people, comprising 53% of the world population, occupy little more than 10% of the worlds’ total land area, which has already led to 60% of the area’s natural land being converted into farmland, grassland, and semi-desert [9]. Consequently, 32% of the total land area is now cultivated in the East Asia monsoon zone [10]. According to the World Bank’s development indicators, the population of South Asia is set to continue to grow by approximately 1.15% per year under current demographic parameters. When coupled with accelerated climate change effects through the East Asia monsoon zone [11,12], this growth risks further degrading ecosystem integrity and functionality and impairing vital ecosystem services, as previously evidenced [13,14,15].

Urban areas typically consist of a matrix of built surfaces with interspersed habitat patches of ‘green spaces’ (such as parks, gardens, remnant habitats) that represent ‘green islands’. These patches can preserve some extent of natural faunistic and floristic assemblages, although species must often adapt their tolerances to enable them to persist [16]. The potential ecosystem benefits of urban green spaces for offsetting carbon emissions, removing air pollutants, and regulating microclimate [17] have been well-established, as has their esthetic appeal and capacity to support community recreational activities [18,19,20]. In comparison, however, their potential to provide wildlife habitat and deliver ecosystem functionality has received less attention. It is therefore important to also consider how effective these urban green spaces are at remediating degraded habitat and to quantify concomitant recovery of ecosystem function [21,22].

Among hierarchical filters determining the community assemblage of remnant species, the complexity of these green spaces interacts with geographic and climatic drivers of regional biodiversity, buffering against species loss or replacement [23]. Seasonality is another important aspect of environmental variability that strongly shapes all aspects of life for organisms living in highly seasonal environments [24]. With respect to temperate ant communities, seasonal generalist species may derive ecological advantages by having a higher maximum critical thermal tolerance and greater thermal plasticity compared with seasonally restricted species [25]. How seasonality mediates habitat filters and influences community assemblages in urban area is particularly important; however, it is inadequately understood, especially for insects.

Ants have a short generation interval and a weak dispersal ability; therefore, ants (including Hymenoptera Formicidae) are taxon sensitive to the ecological impacts of land management [26,27,28]. Habitat openness is generally a key driver of variation in the composition of ant communities, especially because the importance of canopy openness can pervade all aspects of ant life history [29]. Correspondingly, more complex habitats typically harbor more ant species than simpler ones [30,31,32]; however, Ossola et al. [33] report a counter-example from Australia. Furthermore, urban green areas are usually isolated and accompanied by an impermeable surface which often limits the dispersal of ants and plays as a filter on ant communities [34]. Ants are globally dominant faunal group that monopolize up to 25% of the Earth’s tropical faunal biomass [35] and play key ecological roles as soil engineers, predators, nutrient cyclers, and regulators of plant growth and reproduction [26,36,37]. Particular species favor disturbed habitats (defined as conditions under which ecosystem biomass is lost, which is distinct from environmental stress), such as those belonging to the four following functional groups: Dominant Dolichoderinae, Hot-climate Specialists, Opportunists, and Generalised Myrmicinae [29]. In contrast, functional groups that favor closed habitats, such as Cryptic Species, Cold-climate Specialists, and Specialist Predators, are disturbance “losers” [29].

In this study, we compared ant community assemblage composition and species richness across human-modified habitats around Nanchong City, Sichuan Province, in southwestern China. Specifically, we investigated:(i.)If more complex vegetation and less fragmentation resulted in greater ant diversity.(ii.)Whether there were seasonal differences in ant species richness and assemblage composition in relation to any effects of vegetation composition and/or habitat fragmentation.(iii.)Whether and how species richness and assemblage composition sensitivity varied in response to vegetation composition and/or habitat fragmentation between ant functional groups.

## 2. Materials and Methods

### 2.1. Study Area

The study area was located in Nanchong (30°35′–31°51′ N, 105°27′–106°58′ E, 210–850 m) (Figure 1A,B), which is a prefecture-level city in the northeast of Sichuan Province, China, with an area of 12,479.96 km^2^. According to the 2020 census, it was home to over 5.6 million people, for whom the urbanization ratio was 50.2%. Historically, this was a cultivated region interspersed with scattered and severely disturbed woodlands, but it has undergone rapid urbanization since 2003. Currently, it includes a mosaic of secondary woodlands (which are former subtropical evergreen broad-leaved woodlands that have been partially restored with conifers and broad-leaved trees via artificial afforestation, as shown in Figure 1C), man-made cultivated and recreational vegetated areas (Figure 1D), and small croplands that have been cultivated non-intensively (Figure 1E).

Climatically, this region typifies the East Asia monsoon as it is characterized by wet, warm summers and dry, mild winters. During our study period (September 2017 to August 2018) in Nanchong, a maximum temperature of 27.5 °C was recorded in the summer (June–August 2018), accompanied by an average of 111.9 mm monthly rainfall. In spring (March–May 2018), the temperature peaked at 19.1 °C with 125.1 mm average monthly rainfall, while in autumn (September–November 2017), the temperature peaked at 17.4 °C with 83.6 mm average monthly rainfall, and in winter (December 2017–February 2018), the temperature peaked at 7.3 °C with 14.3 mm average monthly rainfall.

In terms of vegetation, the secondary woodlands sampled in our study included various native trees (cypress (*Cupressus funebris*), camphor (*Cinnamomum camphora*), black locust (*Robinia pseudoacacia*), goldenrain tree (*Koelreuteria paniculata*), bishop wood (*Bischofia javanica*), holly (*Ilex chinensis*), etc.), herbs (white cogongrass (*Imperata cylindrica*), dandelion (*Taraxacum mongolicum*), purslane (*Portulaca oleracea*), Chinese motherwort (*Leonurus japonicus*), common carpesium (*Carpesium abrotanoides*), Cretan brake (*Pteris cretica*), adder’s tongue (*Ophioglossum vulgatum*), etc., but it had few shrubs (Figure 1C). These secondary woodlands occurred in scattered patches of less than 4 hectares, and their dominant floristic composition remained relatively consistent among seasons.

Cultivated gardens included planted camphor trees with dwarf lilyturf (*Ophiopogon japonicus*) dominant among the herbaceous vegetation (Figure 1D). Other flora included Bermuda grass (*Cynodon dactylon*), sweet woodruff (*Galium odoratum*), hairy crabgrass (*Digitaria sanguinalis*), Indian aster (*Kalimeris indica*), and occasional waxyleaf privet (*Ligustrum quihoui*) and Chinese privet (*Ligustrum sinense*) shrubs. The dominant floristic composition of gardens also remained relatively consistent among seasons and occurred in scattered patches of less than 2.6 hectares.

Finally, regional croplands comprised a mosaic of small fields cultivated non-intensively (Figure 1E), with crops including maize (*Zea mays*), peanuts (*Arachis hypogaea*), soybeans (*Glycine max*), mung beans (*Vigna radiata*), and oilseed rape (*Brassica napus*) in the spring; maize, peanuts, cowpea (*Vigna unguiculata*), mung beans, and sweet potato (*Ipomoea batatas*) in the summer; sesame (*Sesamum indicum*), pepper (*Capsicum annuum*), cowpea, mung beans, and sweet potato in the autumn; and common beans (*Phaseolus vulgaris*) and peas (*Pisum sativum*) in the winter. Weeds distributed sporadically over croplands included lamb’s-quarters (*Chenopodium album*), horseweed (*Erigeron canadensis*), Chinese plantain (*Plantago asiatica*), and alligator weed (*Alternanthera philoxeroides*). Cropland occurred in dispersed patches of less than 5 hectares.

### 2.2. Ant Sampling and Identification

We conducted pitfall trapping in 1 × 1 m plots to sample ground-dwelling ants. We stratified our sampling regime in a way that was proportionate to the extent each habitat type occurred in the study area. We placed 24 plots in cultivated gardens, 8 in secondary woodlands, and 8 in croplands (Appendix A). This yielded a total of 40 plots (with an inter-plot distance >100 m) (Figure 1). We installed a pitfall trap in the four corners of each plot and surveyed the ants captured for a 48-h period in each month. Plots were permanently located in flat areas that were perpendicular to slopes. Each trap consisted of two 4.5-cm-diameter plastic cups (250 mL) with one cup stacked inside the other. The outer cup remained in the ground, whereas the inner cup and its contents could be removed easily without displacing the entire trap. Traps contained 150 mL of a propylene glycol solution that neither attracts nor repels ants but kills and preserves those captured. Each trap was covered by a small plastic plate attached to three nails to limit solution evaporation and rainfall accumulation. This resulted in a total of 160 trap data points per month (4 × 40 plots) and a study total of 1920 data records (160 × 12 months). We transferred trap contents to 75% ethanol solution to preserve captured ants for further processing [38].

To identify individual ants to species, X.Y.L. and Z.M.Z. used a stereoscopic microscope (Leica M205C) to discern key anatomical features and keyed these out using available literature [39,40,41,42,43] and with reference to the online database AntWeb (http://www.antweb.org (accessed on 12 December 2020)). We excluded worker ants that were too damaged to allow for species identification (194 of 21,707 individuals; 0.89%). We then classified each species into its functional group following Andersen [29]. Specimens were archived at China West Normal University.

### 2.3. Environmental Variables

We assessed six vegetation coverage indices centering a circular survey area of 25 m radius on each plot, including herbaceous vegetation under the canopy (HVUC), herbaceous vegetation outside of the canopy (HVOC), shrub vegetation under the canopy (SVUC), shrub vegetation outside of the canopy (SVOC), bare ground under the canopy (BGUC), bare ground outside of the canopy (BGOC) [44], and two indices of vegetation diversity (i.e. the Shannon–Wiener diversity index and Simpson’s diversity index of vegetation) [45]. We assessed these indices in the summer when vegetation attained its maximum biomass and diversity (Appendix A). Monthly monitoring in other seasons did not reveal any substantial changes among these indices. Simultaneously, we used ArcGIS 10.2 and FRAGSTATS 4.2 to assess how habitat was fragmented by impervious (man-made) surfaces over a 50 m radius centered on each plot by applying six fragmentation indices, including patch density (PD), mean patch size (AREA-MN), landscape division index (DIVISION), mean shape index (SHAPE-MN), effective mesh size (MESH), and splitting index (SPLIT) [46].

### 2.4. Data Analyses

Firstly, we determined the proportion of the 40 sampling sites at which each functional group was detected in terms of the number of individuals and species per group in each season. The presence of ant functional groups was then compared among seasons using a Fisher’s exact test in SPSS statistics software. Only Opportunists and Generalized Myrmicinae occurred in sufficient abundance throughout the year to support our further assemblage model analyses. We tested for spatial autocorrelation for ant species richness using Moran’s I performed in ArcGIS 10.2; however, spatial autocorrelation was not detected.

We used Generalized Linear Mixed Models (GLMM) to assess how ant species richness varied with season and environmental variables. Log-transformed ant species richness was set as the dependent variable, while season and the fourteen environmental variables were set as fixed variables, and the sampling plot was a random intercept (Gaussian error distribution, identity link). Log-transformed ant species richness yielded a better model fit than un-transformed ant species richness. We selected the most parsimonious model according to the Akaike Information Criterion (AIC) value and compared the AIC value of this model with that of the null model (i.e., using only a constant). Marginal and conditional R^2^ values were also calculated to assess how much of the variance was explained by the models [47]. These models were run using the nlme library function in R.

We used a Bioenv analyses (package vegan) to determine which of the fourteen environmental variables best explained ant community composition patterns in this suburban area. We applied a Spearman correlation method with a Bray–Curtis similarity index to the assemblage presence–absence data. We used the Mantel test with 999 permutations to determine whether the results were significant [48,49]. Significance was set at *p* < 0.05 in all analyses.

## 3. Results

### 3.1. Taxonomic Diversity

We caught a total of 21,707 individuals (Table 1). Of these individuals, 21,513 (99.1%) could be identified as belonging to one of 20 species in 12 genera and 4 subfamilies (Table 1). Functionally, eight of these species were Opportunists (12,920 individuals) and six were Generalized Myrmicinae (7935 individuals). Far fewer individuals were caught for other functional groups, including three Cryptic Species (245 individuals), two Specialist Predators (267 individuals), and one Subordinate Camponotini (146 individuals) (Table 1), which was too few to support assemblage model analyses. Opportunists’ numerical presence decreased from a peak of 40/40 plots in summer (8 species) to 28/40 in winter (6 species); Generalized Myrmicinae’s presence declined from 40/40 in summer (6 species) to 23/40 in winter (5 species); Cryptic Species’ presence declined from 17/40 in summer (3 species) to 4/40 in winter (1 species); Specialist Predators’ presence declined from a 21/40 peak in autumn (2 species which were slightly higher than 17/40 in summer) to 1/40 in winter (1 species); and Subordinate Camponotin’s presence declined from 18/40 in summer (1 species) to 2/40 in winter (1 species) (Table 2, Appendix A). Seasonality had a significant influence on the presence of all ant functional groups.

### 3.2. Drivers of Species Richness

The best-fitting model indicated that ant species richness was highest in summer, followed by autumn, spring, and winter (Figure 2A). Ant species richness increased with SVOC and PD and decreased with SPLIT. The fixed effects in the best model explained 61% of the variance, and the random effect explained a further 2% of the variance (the model explained 63% in total) (Table 3).

For Opportunists, species richness was highest in summer, followed by autumn, spring, and winter (Figure 2B). Furthermore, species richness increased with SVOC and PD and decreased with SPLIT. The fixed effects in the best model explained 53% of the variance, and the random effect explained a further 2% of the variance (the model explained 55% in total) (Table 3).

For Generalized Myrmicinae, species richness was highest in summer, followed by spring/autumn and winter (Figure 2C). None of the environmental variables had a significant effect on ant species richness. The fixed effects in the best model explained 52% of the variance, and the random effect explained a further 7% of the variance (the model explained 59% in total) (Table 3).

### 3.3. Drivers of Assemblage Composition

The results of the Bioenv analyses revealed that the environmental variables could explain up to 34.1% of the variance in assemblage composition (Table 4). The best explanatory set of habitat indices and the index with the greatest explanatory value for assemblage composition varied according to the sampling interval (per season or 12 months), but vegetation indices were consistently the most supported predictors of overall ant assemblage composition, specifically for Opportunists, whereas both vegetation and fragmentation indices were supported as predictors for Generalized Myrmicinae.

## 4. Discussion

Our study demonstrates that ant species richness and assemblage composition was affected by three principal factors:(i.)Habitat in relation to vegetation differences within the mosaic of croplands, secondary woodlands, and cultivated gardens where this sort of land-use transformation is ongoing in the East Asia monsoon zone. Greater species richness was mainly contributed to by a greater variety of Opportunist species and occurred in a habitat with higher vegetation coverage and less fragmentation.(ii.)Seasonality. Warm summer conditions were consistently associated with higher ant presence and species richness, which was partly due to higher rates of ant foraging activity [50], and habitat predictors of assemblage composition also varied among seasons.(iii.)Ant functional groups. They differ in species diversity and assemblage composition mediated by vegetation composition and fragmentation.

Our results provide insight into how landscape planning and restoration efforts, such as promoting vegetation coverage and reducing fragmentation, could be used to enhance the diversity of ant species and functional groups, and the important ecological functions that ant communities perform. In general, a better understanding of how to best include ecosystem functionality and not just services [51] is important because subtropical evergreen broad-leaved forests underwent extensive deforestation and urbanization in this region during the latter half of the twentieth century, and it resulted in the loss of over ten thousand square kilometers of core cropland [52].

In terms of best habitat management practice, some croplands in our study were farmed non-intensively with only a low input of pesticides and artificial fertilizers and with cultivation still mainly dependent on manual labor. We recommend that this type of less-intensive cultivation should be perpetuated because it typically supports greater species diversity than more intensive, commercial farming, which retains a high conservation value [53]. In terms of ant assemblages, we found that cropland harbored 18 of the 20 species during the study period. Fortunately, such systems represent 90% of farms worldwide and 60% of total arable land [54]. Indeed, 60% of global studies report that cropland abandonment has a negative impact on biodiversity [55]. We also found that human-modified cultivated and recreational vegetated areas, including cultivated gardens, supported all the 20 species during the study period. Similar benefits have been found for other facets of biodiversity [56], especially in the East Asia monsoon zone; therefore, cultivated gardens can play an important role in maintaining and enhancing ecosystem function, and these benefits should be integrated into future landscape restoration programs. For example, China’s 140 urban parks support high levels of bird diversity, including 495 species, representing 36% of all native species, 49% of total national avian phylogenetic diversity, and 80% of functional diversity [57]. Those woodlands were typical outcomes in this region based on China’s multiple ecological restoration programs [58,59] which supported 19 species during the study period. These policy-driven programs cover over 6.24 million km^2^ of land and have contributed to a strong greening trend and reversion of desertification in China [60]; in contrast, natural restoration has not been so widely adopted [58], although afforestation has proved to be a useful tool to achieve animal conservation outside of protected areas, including for ants [59,61,62].

Regarding specific vegetation coverage indices, in our study, SVOC and PD were both consistently predictors of higher overall ant community species richness, especially for the dominant Opportunists group, whereas SPLIT was consistently associated with lower ant species richness. This aligns with previous studies that found that extensive biodiversity loss typically occurs when land-use systems cause substantial simplification of vegetation structure [63,64].

Weather conditions, especially rainfall, are highly seasonal in the East Asian Monsoon Zone, and seasonality was an important factor in our study as it affected ant species richness and assemblage composition and their responses to habitat effects. The East Asia monsoon zone is characterized by wet, warm summers and dry, mild winters. In summer, large amounts of water vapor from the Pacific and Indian Oceans cause heavy rains over East Asia, whereas in winter, the Siberian high brings predominantly cold air to the continent [65,66]. Generally, ant activity increases in spring or at the beginning of the rainy season and then decreases in the autumn or at the end of the rainy season [67]. Ant activity affects the likelihood of capturing each species, in turn affecting counts of their abundance. The greater activity and the apparent diversity of ants and other insects in warmer months and seasons are due to greater food resource abundance arising from warmer, moister conditions [68]. This seasonal variation in ant activity means it is often necessary to pool data across seasons. Previous studies have shown that the community biodiversity and/or structure of various local taxa are sensitive to seasonality as species richness is positively correlated with contemporary temperature and precipitation, e.g., for birds [69], amphibians [70], planktonic copepods [71], and even microbiome flora [72,73,74].

Turning from extrinsic drivers of ant community composition to the intrinsic qualities of ant species themselves, especially their dispersal ability [75,76,77], we found that the traits of the species involved influenced the way each taxon responded to human landscape modification. Specifically, ant communities in our study were characterized by more species of Opportunists and Generalized Myrmicinae which prefer open habitats and benefit from disturbance [78,79,80]. These functional groups have greater dispersal potential and can compensate some extent of population isolation by distance [81], ultimately affecting assemblage composition. Opportunist species typically tend to dominate in areas with limited ant diversity due to disturbance or environmental stressors. They often colonize disturbed sites using a ‘grab and run’ strategy. In contrast, Generalized Myrmicinae, which include cosmopolitan genera occurring in most habitats, colonize and monopolize sites through their vigorous defense of clumped food resources, followed by rapid recruitment [29]. Such differing tactics between these two functional groups could explain their different species richness and assemblage composition responses, as we observed in relation to habitat effects and/or seasonality. A study looking at the effects arising from prescribed burning in a eucalypt forest in South Australia also noted similar benefits of disturbance for the same ant functional groups [82]. In contrast, ant functional groups that favor closed habitats, such as cryptic species and specialist predators, are typically disturbance “losers” [75,76,77,83]; for example, forest-specialist species underwent a marked decrease in burned plots in the eucalypt study mentioned above.

One limitation of the data collected for our study was that only 20 species were present, mostly due to low diversity in this disturbed region. Careful attention should therefore be given to how to best to encourage dominant and functionally important ant groups across regions of the East Asia monsoon zone that are experiencing land-use type transition and degradation. This will require investigation over more extensive regions and/or in the longer term, including more taxa, and the responses of ant groups that favor closed habitats, such as cryptic species, cold-climate specialists, and specialist predators.

## 5. Conclusions

Our findings concur with earlier reports from other study regions on the relationship between habitat and ant diversity patterns where different human land-cover determined species, phylogenetic, and functional diversity [64,84,85]. Therefore, it is important to manage and mitigate the effects of human-induced impacts on ecosystems, not only in terms of general biodiversity, but especially for key functional invertebrate communities.

Ant community ecology can provide a highly informative bio-indicator [29,86], especially in urban settings [87,88,89]. Previous reports from the coffee agroforests in Ecuador [90], a seasonally dry oak forest in Mexico [91], and savannas in Africa [85] all showed that seasonality affected ant richness, but it did not alter community composition substantially. Based on the importance of vegetation and fragmentation in our study, we recommend that urban habitats should be managed to promote vegetation complexity by enhancing their design beyond simple recreational use to promote ecological benefits [33,56,86]. We propose that habitat corridors should be added and that landscape permeability should be promoted to reduce habitat fragmentation [32,92]. Indeed, under current government policy, the forest coverage in the Chinese portion of the East Asia monsoon zone has increased during the last decade. For example, in the Sichuan province, forest cover has increased from 35.76% in 2014 to 39.63% in 2019, totaling an additional 1,882,100 hectares. Given the significant effects of season and weather conditions, climate change also has the potential to substantially affect ant community composition and dynamics in the East Asia monsoon zone [93,94,95,96]. The influence of these multiple factors suggests the need to more fully integrate functional biodiversity monitoring across taxa, time, and space in order to build ecosystem resilience.

## Figures and Tables

**Figure 1 animals-13-00040-f001:**
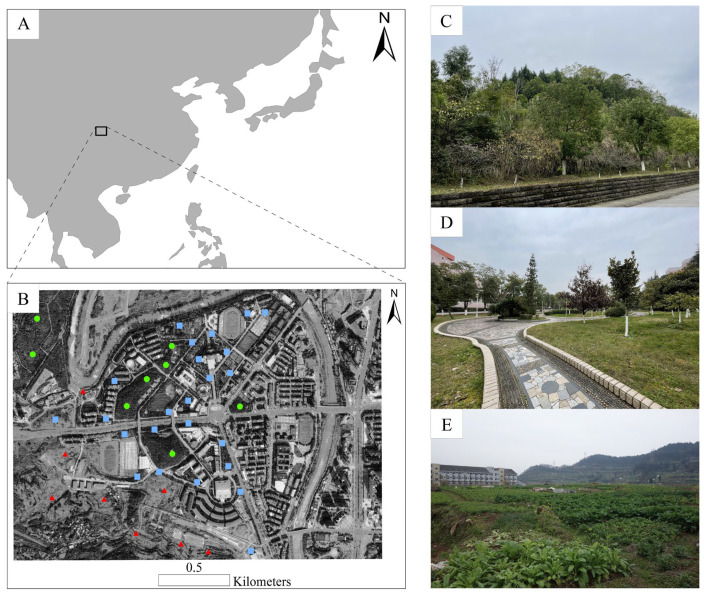
Study location map in China. (**A**) Location of the study area in East Asia; (**B**) sampling locations across the Nanchong metropolitan district; (**C**) secondary woodlands (green dots); (**D**) cultivated gardens (blue squares); and (**E**) croplands (red triangles).

**Figure 2 animals-13-00040-f002:**
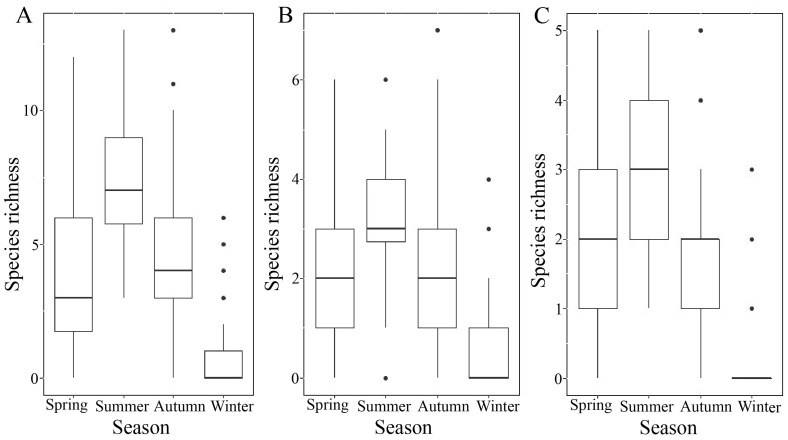
Species richness relative to season. (**A**) The relationship between species richness and season in ant communities; (**B**) species richness of the Opportunist in relation to season; and (**C**) species richness of the Generalized Myrmicinae in relation to season.

**Table 1 animals-13-00040-t001:** Ant species and individual numbers sampled during four seasons (spring, summer, autumn, and winter).

Subfamily	Species *	The Number of Individuals			
		Spring	Summer	Autumn	Winter
Dolichoderinae	*Ochetellus glaber* ^O^	9	92	86	0
Formicinae	*Camponotus japonicus* ^SC^	23	90	30	3
	*Nylanderia bourbonica* ^O^	176	394	269	29
	*Nylanderia flavipes* ^O^	77	189	269	12
	*Plagiolepis manczshurica* ^CS^	53	91	72	7
Myrmicinae	*Aphaenogaster japonica* ^O^	168	386	321	28
	*Crematogaster rogenhoferi* ^GM^	48	41	84	2
	*Crematogaster vagula* ^GM^	3	11	5	0
	*Monomorium pharaonis* ^GM^	227	808	154	9
	*Monomorium chinense* ^GM^	914	1877	343	1
	*Pheidole capellinii* ^GM^	210	454	69	1
	*Pheidole nodus* ^GM^	589	1289	741	55
	*Strumigenys formosa* ^CS^	2	1	4	0
	*Strumigenys hispida* ^CS^	0	8	7	0
	*Tetramorium shensiense* ^O^	124	1029	1407	4
	*Tetramorium tonganum* ^O^	29	135	164	14
	*Tetramorium tsushimae* ^O^	1736	4161	1483	48
Ponerinae	*Leptogenys chinensis* ^SP^	27	111	63	2
	*Leptogenys diminuta* ^SP^	13	6	45	0
	*Odontomachus monticola* ^O^	9	43	29	0

* Abbreviations of ant functional groups include ^O^: Opportunists; ^GM^: Generalized Myrmicinae; ^SC^: Subordinate Camponotini; ^SP^: Specialist Predators; and ^CS^: Cryptic Species.

**Table 2 animals-13-00040-t002:** Proportion of the 40 sites at which each ant group was detected during each season with the number of species detected in parentheses.

Season	Opportunists	Generalized Myrmicinae	Subordinate Camponotini	Specialist Predators	Cryptic Species
Spring	1.00 (8)	0.98 (6)	0.33 (1)	0.20 (2)	0.43 (2)
Summer	1.00 (8)	1.00 (6)	0.45 (1)	0.43 (2)	0.43 (3)
Autumn	1.00 (8)	1.00 (6)	0.33 (1)	0.53 (2)	0.40 (3)
Winter	0.70 (6)	0.58 (5)	0.05 (1)	0.03 (1)	0.10 (1)

**Table 3 animals-13-00040-t003:** Results of generalized linear mixed model analyses with sampling plot set as a random effect for three ant assemblages.

Assemblage	Marginal R^2^	Conditional R^2^	Effect *		Estimate	S.E.	*p*
Ant community (n = 40)	0.61	0.63	Fixed	SVOC	5.00	1.52	0.00
				PD	0.10	0.04	0.03
				SPLIT	−0.66	0.24	0.01
			Random	Sampling plot	S.D. = 0.09		
Opportunists (n = 40)	0.53	0.55	Fixed	SVOC	0.03	1.33	0.01
				PD	0.08	0.04	0.04
				SPLIT	−0.47	0.21	0.03
			Random	Sampling plot	S.D. = 0.08		
Generalized Myrmicinae (n = 40)	0.52	0.59	Fixed	SVOC	3.36	1.73	0.06
				PD	0.04	0.05	0.42
				SPLIT	−0.29	0.27	0.29
			Random	Sampling plot	S.D. = 0.16		

* Abbreviations of environmental variables include SVOC: shrub vegetation outside of the canopy; PD: patch density; and SPLIT: splitting index.

**Table 4 animals-13-00040-t004:** Results of the Bioenv analyses for ant assemblages which show the best explanatory set of environmental variables and the index with the greatest explanatory power.

Assemblage	Sampling Time	Best Explanatory Set of Environmental Variables *		Correlation (Mantel Statistic r)	Index with the Greatest Explanatory Power * (Correlation)	
		Vegetation Indices	Fragmentation Indices		Vegetation Indices	Fragmentation Indices
Ant community	Spring	HVUC, BGUC		0.12 (0.12)	HVUC (0.11)	
	Summer	HVUC, HVOC, BGUC		0.30 (0.30)	HVUC (0.25)	
	Autumn	HVOC, BGUC		0.31 (0.30)	BGUC (0.29)	
	Winter	HVOC, BGUC		0.16 (0.16)	HVOC (0.15)	
	12 months	HVUC, HVOC, BGUC		0.17 (0.17)	HVOC (0.14)	
Opportunists	Spring	HVUC, BGUC		0.14 (0.14)	HVUC (0.13)	
	Summer	HVUC, HVOC		0.32 (0.31)	HVUC (0.28)	
	Autumn	HVUC, HVOC, SVUC, BGUC		0.34 (0.32)	BGUC (0.30)	
	Winter	HVOC, BGUC		0.29 (0.24)	BGUC (0.28)	
	12 months	HVUC, HVOC, BGUC		0.20 (0.20)	HVUC (0.17)	
Generalized Myrmicinae	Spring	HVUC, Simpson’s diversity index		0.13 (0.11)	Simpson’s diversity index (0.11)	
	Summer	HVUC, BGUC	MESH	0.16 (0.15)	BGUC (0.14)	
	Autumn		PD, DIVISION, MESH, SPLIT	0.20 (0.16)		MESH (0.20)
	Winter	HVOC		0.15 (0.15)	HVOC (0.15)	
	12 months	HVUC	SPLIT	0.08 (0.07)		SPLIT (0.07)

* Abbreviations of environmental variables include HVUC: herbaceous vegetation under the canopy; BGUC: bare ground under the canopy; HVOC: herbaceous vegetation outside of the canopy; SVUC: shrub vegetation under the canopy; MESH: effective mesh size; PD: patch density; DIVISION: landscape division index; and SPLIT: splitting index.

## Data Availability

Data supporting reported results can be found in the Appendix A.

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
