# Peer review of "Comparing Ant Assemblages and Functional Groups across Urban Habitats and Seasons in an East Asia Monsoon Climate Area"

_animals, 2022, doi:10.3390/ani13010040_

Round 1

Reviewer 1 Report

My overriding thought about this paper is the potential mis-match with the journal Animals, which appears to be a Veterinary journal or at most a journal about single species zoology, but the paper is about ecology. I will leave it to the journal editors to determine whether the paper is of an appropriate topic.

My next biggest thought is about what it is that the research is aiming to achieve. Line 13 states that it is to “test the effectiveness of ants as a bioindicator of landscape quality.”, and line 21 similarly states “we looked at the quality of 21 sites … using ants as an indicator taxon.” But “quality” is not really defined. Instead, what is presented is differences among ant communities in three different urban habitats. The biggest issue here is that there is no pre-determined measure of high vs low quality, other than the implied metric that highest species diversity must mean higher “quality” habitat. Unfortunately there is no “undisturbed” habitat that can be used as a baseline of “high quality” habitat. This also means that the first goal on line 101 “ if more complex vegetation and fragmentation could reduce the effects of habitat modification on ant communities” really cant be achieved because there is no starting point. Note, that if there is no measure of what something like “high quality habitat” should be, then something else cannot be a “bioindicator” of that status. Instead all there is are differences of a measured taxa in different habitats. This issue is also evident with some other wording, such as on line 276 – cant use words like “responded”, because this is not a manipulative experiment showing before and after. This experiment can only show “differences” among the treatments. Same again on line 299 for “enhanced” and line 300 for “inhibited”. All of these issues can be written out of the paper, but it will change some parts of the paper (Introduction and Discussion) considerably.

Lines 132-147 – typically, a table would be expected to be provided, even as Supplementary Material, giving some quantification of the vegetation present. The description given here is nothing more than a species list. Data collected from the Methods on line 179 to 189 would be appropriate.

Line 253, to state that the environmental variables could explain the variance of assemblage composition is not fully correct. The models at the very best only explained 34% of the variation, meaning that at the very best, 64% (two thirds) remains unexplained.

I am aware of two other related papers with similar research in China which the authors might also find useful to refer to:

Li Q, Hoffmann BD, Lu Z, Chen Y (2017) Ants show that the conservation potential of afforestation efforts in Chinese valley-type savanna is dependent upon the afforestation method. Journal of Insect Conservation 21:621-631

Lu Z, Hoffmann BD, Chen Y (2016) Can reforested and plantation habitats effectively conserve SW China’s ant biodiversity? Biodiversity and Conservation 25:753-770

Author Response

My overriding thought about this paper is the potential mis-match with the journal Animals, which appears to be a Veterinary journal or at most a journal about single species zoology, but the paper is about ecology. I will leave it to the journal editors to determine whether the paper is of an appropriate topic.

R: Many thanks for your kind concerns. Certainly ‘Animals’ has published special issues on veterinary topics, inter alia, and so we understand how you drew this conclusion, but there are also special issues on all kinds of subjects. In general, ‘Animals’ covers a wide range of topics and we have confirmed that our study fits within the broad Aims and Scope of this journal, fitting the category of ecology and conservation, specifically “the abundance and distribution of animals and the relevance to sustainability of the ecosystem”. Therefore, we believe this paper is of an appropriate topic.

My next biggest thought is about what it is that the research is aiming to achieve. Line 13 states that it is to “test the effectiveness of ants as a bioindicator of landscape quality.”, and line 21 similarly states “we looked at the quality of 21 sites … using ants as an indicator taxon.” But “quality” is not really defined. Instead, what is presented is differences among ant communities in three different urban habitats. The biggest issue here is that there is no pre-determined measure of high vs low quality, other than the implied metric that highest species diversity must mean higher “quality” habitat. Unfortunately there is no “undisturbed” habitat that can be used as a baseline of “high quality” habitat. This also means that the first goal on line 101 “ if more complex vegetation and fragmentation could reduce the effects of habitat modification on ant communities” really cant be achieved because there is no starting point. Note, that if there is no measure of what something like “high quality habitat” should be, then something else cannot be a “bioindicator” of that status. Instead all there is are differences of a measured taxa in different habitats. This issue is also evident with some other wording, such as on line 276 – cant use words like “responded”, because this is not a manipulative experiment showing before and after. This experiment can only show “differences” among the treatments. Same again on line 299 for “enhanced” and line 300 for “inhibited”. All of these issues can be written out of the paper, but it will change some parts of the paper (Introduction and Discussion) considerably.

R: We have revised the paper considerably following this comment. The aim of this version turns to focus on “differences” among the treatments, although we retained some mention in Conclusion explaining the general context in which ants can acts as an effective bio-indictor, as proven by the previous research.

Lines 132-147 – typically, a table would be expected to be provided, even as Supplementary Material, giving some quantification of the vegetation present. The description given here is nothing more than a species list. Data collected from the Methods on line 179 to 189 would be appropriate.

R: We have provided an Excel file as Supplementary Material containing detailed habitat quantification and sampling information for each site.

Line 253, to state that the environmental variables could explain the variance of assemblage composition is not fully correct. The models at the very best only explained 34% of the variation, meaning that at the very best, 64% (two thirds) remains unexplained.

R: We have revised the sentence to be “The results of Bioenv analysis revealed that the environmental variables could explain up to 34.1% of the variance in assemblage composition (Table 2)”.

I am aware of two other related papers with similar research in China which the authors might also find useful to refer to:

Li Q, Hoffmann BD, Lu Z, Chen Y (2017) Ants show that the conservation potential of afforestation efforts in Chinese valley-type savanna is dependent upon the afforestation method. Journal of Insect Conservation 21:621-631

Lu Z, Hoffmann BD, Chen Y (2016) Can reforested and plantation habitats effectively conserve SW China’s ant biodiversity? Biodiversity and Conservation 25:753-770

R: Thank you so much for pointing out these two highly relevant papers! We have cited them in this revised version.

Reviewer 2 Report

This paper studied the effects of changes in vegetation and environmental conditions on ant diversity and functional groups. The paper has certain research significance for understanding the effects of human activities on ant population and ants' response to environmental changes. However, the paper still has the following problems, especially in data analysis and presentation.

Line 35-36, add “ant” into keywords

Line 80:, What does “But see [34]” mean?

Line 150-155, this sentence should be revised to be understood well.

Line187-188, what literature did the six fragmentation indices refer to?

Line 220-228, this part of the statement is best represented by a graph or table.

Line233-247, this part is not completely consistent with the information in Figure 2. Figure 2 only provides partial information. Other results can be seen from which charts?

Line253-259, the expression of this part does not well present the results in Table 2, and it is impossible to have a good understanding of the biological significance of the data in Table 2.

Line257, delete and

Line 283, change “km2” into “km2

Author Response

This paper studied the effects of changes in vegetation and environmental conditions on ant diversity and functional groups. The paper has certain research significance for understanding the effects of human activities on ant population and ants' response to environmental changes. However, the paper still has the following problems, especially in data analysis and presentation.

R: Thank you so much for your appreciation of the significance of our paper, and for your constructive suggestions. We have revised the manuscript carefully, following your comments.

Line 35-36, add “ant” into keywords

R: Done.

Line 80: What does “But see [34]” mean?

R: We have revised “but see [34]” to be “however, Ossola et al [33] report a counter-example from Australia”.

Line 150-155, this sentence should be revised to be understood well.

R: For clarity, we have revised “Between September 2017 to August 2018, the average monthly autumn (September - November 2017) temperature in Nanchong was 17.4 °C, with 83.6 mm monthly rainfall; in the winter (December 2017 - February 2018), temperature averaged 7.3 °C with 14.3 mm; in the spring (March - May 2018) conditions averaged 19.1 °C and 125.1 mm; and in the summer (June - August 2018) conditions averaged 27.5 °C and 111.9 mm” to be “During our study period (September 2017 to August 2018) in Nanchong, a maximum temperature of 27.5 °C was recorded in the summer (June - August 2018) accompanied by an average of 111.9 mm monthly rainfall; in spring (March - May 2018) temperature peaked at 19.1 °C with 125.1 mm average monthly rainfall; in autumn (September - November 2017) temperature peaked at 17.4 °C with 83.6 mm average monthly rainfall; and in winter (December 2017 - February 2018) temperature peaked at 7.3 °C with 14.3 mm average monthly rainfall”.

Line187-188, what literature did the six fragmentation indices refer to?

R: Here, we have added the following reference clarifying what the six fragmentation indices refer to: McGarigal, K., Cushman, S. A., & Ene, E. (2012). FRAGSTATS v4: spatial pattern analysis program for categorical and continuous maps. University of Massachusetts, Amherst. http://www.umass.edu/landeco/research/fragstats/fragstats.html.

Line 220-228, this part of the statement is best represented by a graph or table.

R: We have added a table here, presenting the proportion of the 40 sites at which each ant group was detected, by season, with number of species detected in parenthesis.

Line233-247, this part is not completely consistent with the information in Figure 2. Figure 2 only provides partial information. Other results can be seen from which charts?

R: We have added a table to present these other results.

Line253-259, the expression of this part does not well present the results in Table 2, and it is impossible to have a good understanding of the biological significance of the data in Table 2.

R: For clarity, we have revised the table to highlight the biological relevance of these data.

Line257, delete “and”

R: Done.

Line 283, change “km2” into “km2

R: Sorry for our carelessness! Done.

Reviewer 3 Report

The manuscript is overall well written and has merit, however, clarification on resources used for ant identification is needed. The use of a robust taxonomic key or local expert is needed to verify the identification of ants found. 

Author Response

Summary:

Nanchong metropolitan district, and though the results are not necessarily surprising or novel, the approach has merit in its application and advancement of the utilization of bioindicators in such an urban environment. The authors investigated the response of ant species richness and functional group composition in relation to vegetation cover, fragmentation and seasonality and made policy recommendations based off of their findings. The manuscript itself is well written, however, clarification on “quality”of ‘green speces’ would greatly improve the manuscript. Of major concern is the validity of the taxonomic identifications as the references (Bolton 1995, Deyrup (2003), and Antweb) cited by the authors are not sufficient to adequately identify ants to a species level. Deyrup is a checklist of species and provides no taxonomic keys, and antweb can potentially link to keys, further information on what keys were used is needed and suggests that the identification accuracy may be poor which could generate problems in richness and functional diversity estimates, which could impact the results presented here. Overall, the concept is interesting and provides a simple enough strategy to assess urban spaces, which is appealing for management and conservation purposes. I suggest acceptance with major revisions specifically verifying ant identifications with either a local expert or more robust taxonomic keys.

R: Many thanks for your positive and constructive comments on our manuscript.

In this version, we no longer propose that ants serve a direct bio-indicator role in our study and have removed any mention of “green spaces quality” due to the absence of baseline undisturbed habitat. Instead, we present our objective as simply assessing the effect of vegetation indices and fragmentation on ant species richness and assemblage composition.

Following your comments, we consulted more specific taxonomic keys and re-confirmed our species identification. These new taxonomic sources include:

Wu, J.; Wang, C.L. The ants of China; China Forestry Publishing House: Beijing, China, 1995.

Feng, L.A. Taxonomy study on genera Crematogaster and Myrmica from China (Hymenoptera: Formicida: Myrmicinae). Master's Thesis, Guangxi Normal University, Guilin, China, 2007.

Pan, Y.S. Systematic study on the ant genera Pheidole Westwood and Aphaenogaster

Mayr (Hymenoptera: Formincidae: Myrmicinae) in China. Master's Thesis, Guangxi Normal University, Guilin, China, 2007.

Qian, F. Systematic study on the ant genera Tetramorium Mayr and Myrmica Latreille (Hymenoptera: Formicidae: Myrmicinae) from China. Master's Thesis, Guangxi Normal University, Guilin, China, 2008.

For this project, we did not specifically consult with a local ant specialist. However, during the taxonomic identification of over 55,000 ant individuals (82 species) in our other projects, Prof. Zhenghui Xu (an ant expert in Southwest Forestry University, Kunming) confirmed the species of most species involved in this paper. With his assistance and guidance, we are confident in our identification of these local ant species.

General Critiques:

The introduction is organized but would benefit from expanding on a few key points that would make the authors arguments more robust. Specifically, the authors mention the importance of quantifying recovery of ecosystem functions and assessing the quality of habitats using ants, and a clearer link between these two would make it easier for a reader to follow the logic. Secondly, seasonality is one of the stated focuses of the manuscript and so it would be advisable to expand on its role and importance. For the most part, the methods are well presented but analysis may need to address spatial autocorrelation for the map used as well as account for the uneven sampling effort between habitats. As discussed in the general summary, the results are in question until verification of identity accuracy is confirmed. It is also surprising that across 1,920 pitfall traps only 20 species were found when more species have been found in similarly urbanized areas. This may be a result of identification quality already addressed in this review. The discussion needs some reworking to enhance the logical flow the reader is expected to follow.

R: We have revised the paper considerably, particularly by modifying our aim to focus simply on “differences” in ant community composition among the treatments. As general context, however, in the Conclusion, we retain some mention of the potential for ants to serve as bio-indicators, as proven by the previous research.

We have added the following content to the Introduction, better explaining the role and importance of seasonality: “With respect to temperate ant communities, seasonal generalist species may derive ecological advantages through having a higher maximum critical thermal tolerance and greater thermal plasticity, compared to seasonally restricted species [25]”.

To address spatial autocorrelation for the map, we have added the following content into section 2.4 Data Analyses: “We tested for spatial autocorrelation for ant species richness using Moran's I, performed in ArcGIS 10.2; however, spatial autocorrelation was not detected”.

Our paper only uses vegetation indices and fragmentation indices as the potential predictors of ant community composition; it does not use habitat types as a predictor, and so there should be no issues relating to uneven sampling effort between habitats. To avoid misleading readers in this regard, our revision retains “habitat types” only to present the type of landscape in the studied area.

The low species richness in our study was genuine and may be attributable to: i) the high degree of disturbance in the study area; 2) the relatively small range of the study area, at just c. 2,200 meters in diameter; 3) only 40 sites with 160 pitfall traps being set per month; or 4) because ants have weak dispersal ability, limiting the range of species living in this area.

We have considerably revised the Discussion to enhance its logical flow.

Line by line

55-59- Sentence could use some minor edits to help clarify the point the author is trying to make.

R: We have revised this sentence to read: “The potential ecosystem services benefits of urban green spaces, for offsetting carbon emissions, removing air pollutants and regulating microclimate [17], have been well-established, as has their esthetic appeal and capacity to support community recreational activities [18-20]. In comparison, however, their potential to provide wildlife habitat and deliver ecosystem functionality has received less attention”.

72-73 No need for examples

R: This paragraph has been totally deleted, because this revision no longer refers to the role of ants as bio-indicators.

75-76- This transition between paragraphs is rushed and would benefit from rewording or expanding. Also unclear on what the authors mean by bio-indicator.

R: This paragraph has been totally deleted, as above.

101- Are you expecting more vegetation cover and more fragmentation to reduce the effects of habitat modifications? Also unclear what complex fragmentation means and how this would lead to reduced effects. Further information on fragmentation and its effects on the ant community would be welcomed.

R: We have revised “if more complex vegetation and fragmentation could reduce the effects of habitat modification on ant communities” to read: “if more complex vegetation and less fragmentation resulted in greater ant diversity”.

For further information on vegetation and fragmentation and how these affect the ant community, we have added the following two sentences into the Introduction: “Habitat openness is generally a key driver of variation in the composition of ant communities, especially that the importance of canopy openness can pervade all aspects of ant life history [29]” and “Furthermore, urban green areas are usually isolated and accompanied by impermeable surface, which often limits the dispersal of ants and plays as a filter on ant communities [34]”.

149-155- Paragraph should be placed earlier, before the introduction of different habitat types

R: Done.

158-160- What is the reason to have three times more sites in one habitat type than the other two?

R: This paper does not aim to assess the effects of habitat types on ant community responses. To cover the range of species in the studied area optimally, “we stratified our sampling regime in a way proportionate to the extent each habitat type occurred in the study area”.

172-173- References not useful for accurate identification

R: We have revised “We identified individuals to the species level following Bolton (1995), Deyrup (2003) and AntWeb (http://www.antweb.org) [40,41]” to read “To identify individual ants to species, X.Y.L. and Z.M.Z. used a stereoscopic microscope (Leica M205C) to discern key anatomical features and keyed these out using available literature [39-43] and with reference to the online database AntWeb (http://www.antweb.org)”.

181-183- Does the vegetation change throughout the season and was that accounted for?

R: Yes, there are seasonal patterns of changing vegetation indices. However, i) the dominant floristic compositions of secondary woodlands and the cultivated gardens remained relatively consistent among seasons; and 2) non-intensive cultivation was perpetuated in the croplands, although the crops grown varied with seasons. For consistency, we assessed vegetation indices in the summer, when vegetation generally achieved maximal biomass and diversity. Monthly monitoring in other seasons did not reveal any substantial changes in these indices. We have provided these details in this revision as follows: “We assessed these indices in the summer when vegetation attained its maximum biomass and diversity. Monthly monitoring in other seasons did not reveal any substantial changes among these indices”.

191- Rewording may improve reader comprehension.

R: We have re-phrased this section to improve clarity as follows: “Firstly, we determined the proportion of the 40 sampling sites at which each functional group was detected, in terms of number of individuals and species per group in each season”.

276-277- Evidence for this claim needs to be provided or made clearer in the results beyond that of table 1.

R: We have changed this to read “(iii) ant functional groups - which differ in species diversity and assemblage composition mediated by vegetation composition and fragmentation”. This claim was arrived at based on the Results given in sections 3.2 and 3.3. Furthermore, we revised Table 1 to present the individual number of each species across seasons, rather than by habitat.

280- Could use more discussion after the three points demonstrated on what this means for ecosystems and how the functional groups present or absent informs that interpretation.

R: We have included some further discussion on implications for the ecosystem and how the functional groups present or absent informs that interpretation as follows: “Our results provide insight into how landscape planning and restoration efforts, such as promoting vegetation coverage and reducing fragmentation, could be managed to enhance the important ecological functions ant communities perform, which means the presence of more species and functional groups. In general, a better understanding of how best to include ecosystem functionality, and not just services [51] is important because, in this region, subtropical evergreen broad-leaved forests underwent extensive deforestation and urbanization during the latter half of the twentieth century that resulted in the loss of over ten thousand square kilometers of core cropland [52]”.

281-297- May be beneficial to move this paragraph later in the discussion

R: This paragraph has been revised and relocated as the discussion for the secondary woodlands, better fitting the continuity of discussion.

304- May need to rework this paragraph in order to more clearly make the point of the role of ant dispersal in this system.

R: We have re-worked this paragraph, and hope that role of ant dispersal in contributing to the patterns we found is now clearer.

Figure 1 B. Scale of map would be nice to include

R: Done.

Table 1:

Monomorium pharaonic should be spelled Monomorium pharaonic

Strunigenys Formosa should be Strumigenys Formosa

R: Sorry for our carelessness! Done.

Round 2

Reviewer 1 Report

.

Author Response

R: There are no comments or suggestions from this reviewer, and so we are delighted that our revision was satisfactory and met with their approval.

Reviewer 2 Report

The author has made detailed revisions to the manuscript, especially the more in-depth analysis of the data and the addition of tables. But there are still two small problems.

Line 262, on the horizontal axis in Figure 2, the order of seasons should be spring, summer, autumn and winter.

Line 281, In Table 4, the data is reserved for two decimal places.

Author Response

The author has made detailed revisions to the manuscript, especially the more in-depth analysis of the data and the addition of tables. But there are still two small problems.

R: Thank you so much for your positive comments! We have addressed the following two problems.

Line 262, on the horizontal axis in Figure 2, the order of seasons should be spring, summer, autumn and winter.

R: Done

Line 281, In Table 4, the data is reserved for two decimal places.

R: Done.

Reviewer 3 Report

The authors did a good job addressing the concerns raised and increasing the logical flow of the paper. There are no glaring issues in this draft, though line 94-95 needs to be reworked to make it more comprehensible to the reader.

Author Response

The authors did a good job addressing the concerns raised and increasing the logical flow of the paper. There are no glaring issues in this draft, though line 94-95 needs to be reworked to make it more comprehensible to the reader.

R: Thank you so much for your positive comments! We have revised “whether and how there were any seasonal differences in ant presence and among habitat effects on ant species richness and assemblage composition” to read “whether there were seasonal differences in ant species richness and assemblage composition, in relation to any effects of vegetation composition and / or habitat fragmentation”.